# Evaluation of the delivery of the family nurse partnership programme in Scotland during the COVID-19 pandemic

Lawrence Doi[ID][1]*, Kathleen Morrison[1], Emmanuel Kwadwo Anago[ID][1,2], Thomas Hughes[1,3]

1 Scottish Collaboration for Public Health Research and Policy, School of Health in Social Science, University of Edinburgh, Edinburgh, Scotland, 2 School of Nursing and Midwifery, Presbyterian University, Ghana, 3 eSafety Commissioner, Level 32 Melbourne Central Tower, Melbourne, Victoria, Australia

* larry.doi@ed.ac.uk

## Abstract

### Background

The Family Nurse Partnership (FNP) is an intensive and structured person-centred home-visiting programme delivered by specially trained nurses, who offer support services to first-time young mothers. The COVID-19 pandemic prompted the quick adoption of telehealth within the FNP, as healthcare services moved rapidly to implement remote delivery systems in line with infection control measures. The aim of this study was 1) to understand the features of telehealth employed to deliver the FNP programme during COVID-19 in Scotland; 2) to examine how FNP nurses and clients responded to the delivery of FNP through telehealth; 3) to evaluate the challenges of delivering the FNP through telehealth during COVID-19 and its implications for future delivery of the programme.

### Methods

The study employed a mixed-methods parallel design, where qualitative (one-to-one interviews and focus groups) and quantitative (survey) data were collected and analysed concurrently. Thirty-one family nurses took part in the focus groups and one-to-one interviews and a further 90 responded to the online survey. Fifteen FNP clients participated in one-to-one interviews. Interview data were analysed using thematic analysis and survey data were analysed by descriptive analysis.

### Results

Family nurses combined both home visiting and remote delivery such as phone calls, SMS text messaging, emails, video calls to deliver the programme. Family nurses felt well equipped and supported to conduct their work remotely. Clients, particularly those who became isolated during COVID-19, overwhelmingly acknowledged this

**Data availability statement:** Ethical restriction by the Public Benefit and Privacy Panel for Health and Social Care (HSC-PBPP) in Scotland prohibits the authors from making the minimal data set publicly available because data contain potentially identifiable information which may compromise patient and staff privacy. However, request to access the data from the researchers could be made through the Ethics Committee via phs.PBPP@phs.scot".

**Funding:** The evaluation was funded by a grant from the Scottish Government, with reference number: C15968.

**Competing interests:** The authors have declared that no competing interests exist.

support and felt their family nurses provided stability, advice and care. However, both family nurses and clients found the rapid move to remote delivery challenging, because it affected both recruitment of clients with complex vulnerabilities to the programme and therapeutic relationship building. Nevertheless, 42% of family nurse respondents in the survey indicated that they would prefer mixed-mode delivery of face-to-face and telehealth as part of future FNP programme delivery.

## Conclusion

Despite the challenges of delivering the programme remotely during COVID-19, telehealth has the potential to play a valuable role in post COVID-19 FNP programme delivery. A hybrid delivery approach could be appropriate in certain instances, for example clients not deemed to have complex vulnerabilities or those living in remote locations. Future studies could robustly examine the impact of the quality of modes of FNP delivery, for instance home visiting, telehealth and hybrid delivery and how these influence outcomes across different client groups. An economic evaluation of the value for money of different modes of delivery could also be insightful for decision makers.

---

## Introduction

The Family Nurse Partnership (FNP) is a programme specifically designed to deliver support services to first-time young mothers and their children through a structured one-to-one home visiting format [1]. Originally designed and developed in the United States as the Nurse-Family Partnership, the license-based programme provided preventive services for low-income first-time young mothers [2]. At the core of the delivery model were specially trained nurses who delivered the intensive person-centred home-visiting programme [2]. The programme seeks to achieve three key objectives: 1) enhancing prenatal health and pregnancy outcomes in young women; 2) enhancing the overall health and development of children by empowering and assisting young parents to deliver competent childcare; and 3) improving the life of these young parents by providing services that encompass family planning, education and career [2]. Several other countries have since adopted the FNP into their health systems such as Canada, the Netherlands, and the United Kingdom (UK) (excluding Wales) [3–5].

Since the inception of the FNP programme in the UK, several studies have been conducted across England on the impact of the programme. Sanders and colleagues used a focus group discussion to explore the views of key healthcare professionals (family nurses, health visitors and midwives) regarding the provision of the FNP programme [1]. The findings revealed that although the family nurses showed confidence and optimism in the effectiveness of the FNP programme, they encountered challenges in meeting programme targets and indicated that the primary objectives of the programme did not mirror the priorities of the community or clients [1]. A

qualitative case study of 18 low-income first-time young mothers in Canada regarding the FNP programme revealed participants' positive experiences concerning the programme [6]. The study identified that the FNP nurses were competently able to create an atmosphere that fostered therapeutic relationships with the participants which in turn led them to achieve positive outcomes [6].

A study in the Netherlands, where the FNP programme is termed 'VoorZorg' or the Nurse-Family Partnership (N-FP), assessed the impact of the programme on child maltreatment and development [3]. The authors designed a randomised controlled trial to compare regular care against the N-FP home visit programme. The study included 460 participants who were disadvantaged women, aged 26 years or under, and pregnant with their first child. The study began from pregnancy and childbirth through until the children were two years old. The primary outcome measure of the study was the presence of reports concerning a child from a Child Protecting Services (CPS) agency. The home environment and child behaviour were stated as secondary outcomes. The authors found that 19% of children in the control group had a CPS report compared with 11% of children in the intervention group three years after birth, which was significantly lower [3]. Children in the intervention group also showed significant improvements in internalising behaviour to those in the control group [3].

## The Family Nurse Partnership (FNP) in Scotland

The FNP was introduced to Scotland in 2009. The FNP in Scotland is provided by qualified and skilled nurses and midwives who have successfully completed prerequisite master's level training. Since its introduction, the FNP has been extended to more than 10,000 families across Scotland [7,8]. While the FNP was originally designed to provide support services to first-time mothers aged 19 years and below, this has recently been expanded to young parents aged up to 24 years old in Scotland [8].

The COVID-19 pandemic placed a significant burden on the health and economy of many countries across the globe, including Scotland [7]. A report of the Scottish Government during the COVID-19 crisis stated that "the COVID-19 pandemic has brought a renewed focus on the specific vulnerability of the client group who receive FNP and it is imperative to recognise the essential role of FNP in response to the pandemic" [9]. The burden of the COVID-19 pandemic exacerbated the already existing difficulties and disparities faced by vulnerable and minority populations [10]. Additionally, a nationwide report in the UK underscoring the effect of the COVID-19 pandemic and the first UK lockdown on children (0–2 years old), showed that young children were indirectly affected by the COVID-19 crisis by being more susceptible and vulnerable to a broad range of 'hidden harms' [11]. These 'hidden harms' comprised factors such as poor parental mental health, traumatic experiences, invisibility to professionals [11]. Therefore, many service providers and policy officials judged that the FNP programme was an important service to help mitigate these harms during the COVID-19 pandemic.

The COVID-19 pandemic brought a focus on telehealth as healthcare services moved rapidly to implement remote delivery systems in line with infection control measures. A national clinical guidance was produced by the Scottish Government to guide nursing and allied health professionals and community health staff during the COVID-19 pandemic [12]. Fundamental and essential changes occurred to enable and assist the delivery of regular and continuous care alongside crucial services, where possible. Regarding the FNP, this involved a predominant shift to remote service delivery using technology such as telephone calls, SMS messaging and video calls via platforms such as 'Attend Anywhere' or 'Near Me' [9]. This supported clients to maintain regular contact with their family nurses. Home visiting was provided for vital circumstances only [9].

## Rationale and aims of study

Some challenges of telehealth usage during the COVID-19 pandemic have been captured by a systematic review, which included papers from 64 studies across 18 countries [13]. The review found that the most common challenges were technology acceptance and adoption, concerns about the accurateness of subjective patients' assessment and technical issues with technology [13].

Further, a study has revealed that young mothers in the UK are less likely to engage with digital health and online support sources, relying instead on trusted interpersonal sources and community-based support that adequately address the complexity of their needs [14]. Considering the FNP programme targets young mothers and the essential role telehealth played in the delivery of the FNP programme in response to the COVID-19 pandemic, it is vital to gain a thorough understanding from the perspectives of FNP nurses and their clients to better characterise the implications of the service adaptation for future programme delivery. This study, therefore, addressed the following aims:

1) to understand the features of telehealth employed to deliver the FNP programme during COVID-19 in Scotland; 2) to examine how FNP nurses and clients responded to the delivery of FNP through telehealth; 3) to evaluate the challenges of delivering the FNP through telehealth during COVID-19 and its implications for future delivery of the programme.

## Methods

### Design, setting and context

This evaluation employed a mixed-methods parallel (convergent) design. Firstly, qualitative (interviews and focus groups) and quantitative data (survey) were collected concurrently. Secondly, we independently analysed qualitative and quantitative data using appropriate methods (see data analysis section below for detail). Finally, data integration occurred by comparing and contrasting the results from both analyses. The approach facilitated a more comprehensive and nuanced understanding of the research problem because the qualitative findings provided in-depth explanation to the quantitative results.

The study was conducted in the eleven (of a total of fourteen) Scottish Health Boards currently running the FNP programme. However, Health Boards were not identified in this paper to protect the identity of participants. Data collection was carried out from January 2021 to March 2021, and this was during a period where a range of public health measures and restrictions were in place to mitigate the COVID-19 pandemic. In the UK, a strict national lockdown was in place from 24th March to 28th May 2020. Lockdown restrictions were gradually eased in phases until regional level restrictions were implemented across Scotland on 1st November 2020. However, from January 2021 to April 2021 a second national lockdown was introduced.

### Sample

Thirty-one family nurses took part in one-to-one interviews (n = 23) and focus groups (n = 8). Fifteen clients were involved in one-to-one interviews: six clients were enrolled onto the FNP programme prior to the COVID-19 pandemic, nine clients were enrolled after the introduction of COVID-19 restrictions. As of 31st March 2021 there were 2,909 active FNP clients in Scotland [15]. All family nurses across Scotland were invited to complete an online survey and ninety of them completed it at eligible response rate of 41%.

### Recruitment and data collection

Due to COVID-19 restrictions in place at the time of the research, all data were collected remotely using telephone and digital methods, including video calls and online surveys. Throughout data collection, participants were asked to reflect on their experiences of delivering and receiving the FNP service following the COVID-19 outbreak in March 2020 up until the point of data collection in January 2021 to March 2021.

### Qualitative interviews and focus groups

Family nurses were recruited via nurse managers who distributed study invitation letters and information sheets to their staff by email. Those interested in taking part in either a one-to-one interview or focus group completed informed consent via Qualtrics using an online data form.

FNP clients were recruited by their family nurses. Family nurses were provided with the study recruitment materials containing participant and study information sheets, which were shared with clients in various formats (i.e., text message, email, or verbal conversation). Clients interested in the study contacted the research team directly by completing an online Qualtrics data form to provide informed consent. Options were also provided to contact the research team via email with any questions prior to participating. The online form allowed clients the option of choosing a preferred time and day for a telephone interview. We also gave clients the option to request a female researcher or additional support from their family nurse, if required.

Interviews were conducted mainly by telephone. In few instances Microsoft Teams was used. All focus groups were conducted via the Microsoft Teams video platform. All interviews and focus groups were recorded using an encrypted digital audio recorder and then transcribed verbatim. All transcripts were anonymised and pseudonymised prior to analysis. Each client received a £20 high-street voucher as a thank you for their participation.

The interviews and focus groups were conducted by KM, TH, and LD. KM worked as a Research Associate, and holds an Masters Degree in Public Health and has extensive experience of conducting evaluations of home visiting programmes. LD is a Senior Lecturer in Applied Public Health and holds a PhD in Public Health. TH worked as a Research Assistant and holds an MSc in Health Policy. All members of the research team involved in undertaking the interviews had received training and had experience in qualitative methods and interview etiquette. Both TH and LD are male, while KM is female. None of the researchers had prior relationships with any of the participants in the study.

## Survey of family nurses

Family nurses were invited to take part in the survey through their local nurse managers, who distributed links to the Qualtrics survey via email. The survey questionnaire was developed using existing FNP programme documents and findings from a rapid review [5], therefore no validated instrument was used. For participants to access the survey questionnaire, they had to first access the participant information sheet and complete a consent form. Informed consent was received via Qualtrics using an online data form. Participation in the survey was anonymous.

## Data analysis

**Qualitative data.** The qualitative data were analysed using thematic analysis [16]. Following familiarisation with the data, we developed initial deductive coding frameworks to code staff and client transcripts using categories and sub-categories drawn from existing FNP literature, FNP programme documents and stakeholder discussion. The coding frameworks were tested independently by authors (KM, TH) on some transcripts to check consistency. Once the coding frameworks were agreed, all transcripts were coded and using NVivo software. New or emergent codes were discussed amongst all authors and included if relevant. A number of transcripts were periodically crosschecked and double coded by another member of the authorship team to further ensure consistency. No notable disagreements in coding were identified and minor differences were resolved through discussion. Similar codes were categorised and grouped into themes. Themes were iteratively discussed, refined and named appropriately by the research team [16].

**Survey data.** Aided by MS Excel, data from the survey were analysed descriptively, using graphs and tables to present findings by proportions. Due to moderate sample sizes, data are presented at an aggregated level across Health Boards in order to preserve anonymity. Text field data was analysed thematically by two of the authors (KM, TH), with another author (LD) cross-checking the themes to ensure consistency.

In line with mixed-methods parallel design, data from the qualitative research and survey were triangulated, by comparing and presenting the results from both analyses to identify any convergence or divergence across the data, providing further credibility and validity to the findings. For ease of navigation, the findings are reported based on themes or topics.

## Ethics and approvals

This research was approved by the School of Health in Social Science Research Ethics Committee, University of Edinburgh and Tier 1 approval was also gained from the Public Benefit and Privacy Panel for Health and Social Care (HSC-PBPP) in Scotland. Each participant completed a written informed consent prior to taking part in the study.

Ethical restriction by the Public Benefit and Privacy Panel for Health and Social Care (HSC-PBPP) in Scotland prohibits the authors from making the minimal data set publicly available because data contain potentially identifiable information which may compromise patient and staff privacy. However, request to access the data from the researchers could be made through the Ethics Committee via phs.PBPP@phs.scot.

## Results

### Sample

Thirty-one family nurses took part in the qualitative research and 90 responded to the online survey. Fifteen clients participated in one-to-one interviews. The results are presented thematically.

### Prior experiences of FNP service provision via telehealth

The use of a range of telehealth modes in the FNP delivery prior to the COVID-19 pandemic was infrequent. In the qualitative study many family nurses reported that they had never used telehealth modes in delivering FNP services prior to the pandemic. However, in the survey, 92% reported using telephone calls. Only 9% and 7% of family nurses reported previously using 'Attend Anywhere' and 'Near Me' platforms respectively, while 11% reported using other video call platforms with clients, see Table 1. Aside from telephone use, low usage of video call platforms by family nurses prior to COVID-19 is in line with existing evidence that demonstrated that despite its potential benefits for patients and healthcare providers, telehealth was underutilised and understudied prior to COVID-19 [17].

### Response to COVID-19 outbreak and restrictions

Several family nurses mentioned that the initial six weeks subsequent to the UK-based outbreak of the COVID-19 and the onset of national lockdown restrictions brought the most uncertainty concerning the FNP service provision. The closure of offices and workplaces in March 2020 led to an abrupt shift to home working for FNP nurses and changes were implemented rapidly across the service. Swift adaptation was required to deliver FNP in accordance with newly introduced public health measures and restrictions while using locally available technologies and resources.

> "So I very quickly had to get my laptop, collect a pile of paperwork and head home, as we were advised to do, and it took me quite a bit of time to work out how to do remote working and get all that set up, and I just found it incredibly stressful, it was like learning a whole new job." [Family Nurse, E]

**Table 1. Previous experiences of using different telehealth modes to deliver FNP prior to the COVID-19 pandemic.**

| Telehealth Mode | Yes – previous experience | No– previous experience | Don't know |
|---|---|---|---|
| Telephone calls | 92% | 8% | 0% |
| WhatsApp | 31% | 69% | 0% |
| Attend Anywhere | 9% | 91% | 0% |
| Near Me | 7% | 92% | 1% |
| Video Call (other) | 11% | 89% | 0% |

*"It felt like overnight it was, like, that's it. You know, no home visit. We need to be really careful. It's lockdown." [Family Nurse, N]*

## Modes of delivery

Family nurses were predominantly delivering the service remotely, but in vital circumstances home visiting was possible. Home visiting included face-to-face and outdoor meetings with clients, while remote delivery included phone calls, SMS 'text' messaging, emails, video calls, WhatsApp and other mobile media apps. Although various communication formats were suited to different forms of contact or programme delivery more than others, a significant finding was that having a range of communication options was highly beneficial for family nurses. Text messaging, telephone calls and video calls were among the most frequently used telehealth modes. See Table 2.

## Home visiting

The ability to offer home visits during the COVID-19 outbreak was critical to family nurses as it allowed them to offer timely and vital support to many vulnerable and at-risk clients during this time. All surveyed staff (100%) reported delivering home visits during the COVID-19 outbreak. When a home visit was offered to clients, 41% of family nurses reported that offers of home visits were always taken up while 51% indicated that these were taken up most of the time. Barely half of clients not taking up home visit offers perhaps reflects the client group that the FNP programmes often serve. A vulnerable client group often sceptical about health professionals and reluctant to engage with their services [18]. The proportion of caseloads who were offered home visits from March 2020 to March 2021 varied between family nurses as shown in Fig 1. Twenty-nine percent of family nurses offered home visits to 100% of their caseloads; 15% of family nurses offered home visits to 75–99% of their caseload; 20% of family nurses offered visits to 50–74% of their caseload; and 22% of family nurses offered home visits to 25–49% of their caseload.

Crucial home visits were typically conducted by family nurses for developmental assessments, new-born visits, and child protection reasons. Clients with mental health conditions and those at risk of domestic violence were also prioritised for home visits. For non-English speaking clients, visits to their homes were undertaken by family nurses while connecting with their interpreters remotely.

*"So in COVID we've obviously had to be directed by Scottish Government to which visits we can do face to face, in their communication with us which visits we could do face to face and which visits have to be done on a digital pathway. However, in all of this we have had...we're qualified nurses so we have had freedom to be able to assess ourselves whether a face-to-face visit is required. And predominantly I would say that would be for domestic abuse or for child*

**Table 2. Frequency of use of service delivery modes reported by family nurses.**

| Mode of Service Delivery | Always | Most of the time | About half the time | Sometimes | Never |
|---|---|---|---|---|---|
| Telephone calls | 5% | 23% | 33% | 38% | 1% |
| WhatsApp | 3% | 8% | 3% | 31% | 56% |
| Mobile apps | 0% | 0% | 3% | 18% | 79% |
| Attend Anywhere | 1% | 12% | 22% | 24% | 40% |
| Near Me | 1% | 18% | 24% | 36% | 21% |
| Video Call (other) | 0% | 9% | 8% | 18% | 65% |
| Text messaging | 21% | 23% | 11% | 27% | 17% |
| Home visits | 1% | 19% | 34% | 45% | 1% |
| Home visits | 1% | 19% | 34% | 45% | 1% |

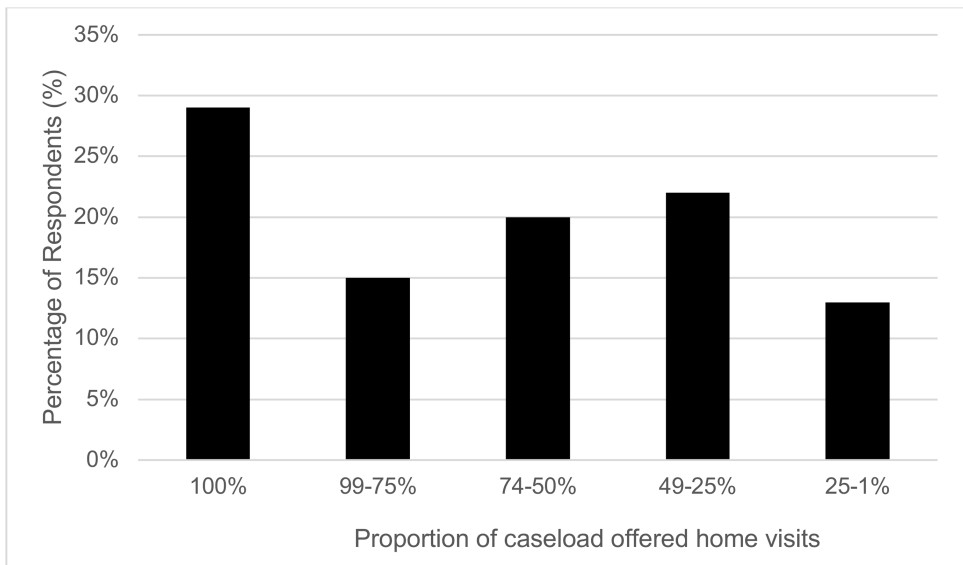

**Fig 1. Reported proportion of FNP client caseloads offered home visits during COVID- 19 outbreak, from March 2020–2021.**

*protection. [...] We've had to work very carefully with Scottish Government from a FNP point of view [and] with our health board as well, but we've had to keep our clients and our nurses safe." [Family Nurse, N]*

Often clients understood COVID-19 risks when being offered a home visit. Most clients felt safe during home visits and noted that they trusted their family nurses as health professionals to appropriately use Personal Protective Equipment (PPE) and mitigate transmission risks.

*"With the PPE and stuff, we were all well protected, she had her mask and her gloves and her apron on at all times, she never came in the house without it. And I knew at any point if she felt she wasn't well or anyone in this house wasn't well, that we wouldn't allow her to come in, so we felt safe with her in the house." [Client (W), enrolled during COVID-19]*

Where home visits were offered, most clients were reported to take these up. However, in some instances, clients declined home visits due to reasons such as being clinically vulnerable or at higher risk of serious illness if infected with COVID-19; having a 'shielding' family member; being afraid of COVID-19 infection risks; or having to self-isolate as a result of COVID-19 exposure.

Similarly, some family nurses also reported feeling afraid or anxious about contracting COVID-19 during in-person visits. Some family nurses perceived that it was not always safe to enter the clients' home due to large numbers of people sharing a household, despite taking precautions such as wearing PPE.

*"They're also going to home visits that sometimes, despite every best effort, they're doing their risk assessment, they're trying to keep themselves safe, they're going in and having their PPE on, but they go in and the house is really full with other family members. And they're thinking to themselves, 'are they all following the rules ... I'm in the house and putting myself out there, even though I'm trying to follow the rules myself.' And that for them at times has been scary." [Family Nurse, W]*

### Remote delivery

**SMS (text messaging).**  Text messaging was commonly used between clients and family nurses for scheduling appointments, providing updates or reminders, and communicating about concerns relating to the client or their child. Text messaging was described as a longstanding method of communication in the programme (pre-COVID-19) and was perceived as being very accessible format for both clients and family nurses to communicate in between scheduled contacts.

*"[My family nurse has] given me her number so if I ever felt the need to I could just text her if I had any concerns but like that sort of calling, that's usually like once every four weeks or so." [Client (W), enrolled during COVID-19]*

**Telephone calls.**  Several family nurses described using telephone calls to some extent to deliver FNP remotely. Family nurses reported that the majority of their remote contacts with clients were conducted over the phone. Telephone calls were identified as a preferred option for many clients despite the option of video calls being available.

*"My clients, at the moment, most of them want phone calls rather than Near Me." [Family Nurse, E]*

**Video calls: NHS attend anywhere & NHS Near Me.**  NHS Attend Anywhere and NHS Near Me were frequently cited as a key mode of approved video communication with clients. Most clients and family nurses perceived the platforms as being intuitive and easy to use.

*"It's really easy to use Near Me, it's really easy to use and the way it's set out and when your client reads it, it's really welcoming for the clients. [It names] the four nurses in our team… and I think that's really nice, so the client knows they're in the right place. Actually, when you text them your appointment and you put the link in it automatically comes up with a link on their phone, so they just click onto it on their phone if they've got an iPhone or a smartphone. That is so easy for them and I really like that. So, I think the system is really good. I think the system is really smart… It's a really efficient system." [Family Nurse, W]*

*"So, technology comes quite easy for me, I know it doesn't to a lot of people. So, I can attend anywhere and stuff that's used to Zoom and things like that, all comes second nature to me because just now a lot of my life is online." [Client (W), enrolled during COVID-19]*

Many family nurses also reported that clients felt uncomfortable and anxious about engaging in video calls compared to other forms of contact.

*"They're starting to get used to it, but it's a new way of working, even for clients, and I think some struggle, and also, obviously issues with IT, and Wi-Fi, data, you know, all the other things that go along with that." [Family Nurse, E]*

Some family nurses mentioned that their teams had been asked to nominate local champions for the use of NHS Near Me/NHS Attend Anywhere. This was described as a useful and positive link to support teams troubleshoot and resolve issues with the platforms.

**Social Media and other communication platforms.**  WhatsApp was considered to be a widely available, trusted and accessible to clients which enabled them to conduct phone calls, video calls and send messages to their family nurses. Family nurses felt this mode of communication was particularly beneficial when engaging with minority groups and for being able to send documents and resources to clients.

*"And for me, there felt like a lot of pressure to move to Attend Anywhere from top down. When actually I've always felt, and I still maintain that using a WhatsApp video call was more accessible to lots of the clients. Because as a service*

*we went to them. Before COVID we went to them. We made it easy, we went to their house, we met them where they wanted to meet us." [Family Nurse, W]*

Family nurses also used websites resources and NHS YouTube clips for sharing information and conducting demonstrations with clients. They felt it was sometimes more effective than providing clients with text heavy resources.

*"I've used a lot of NHS YouTube clips [for] a few clients in the pregnancy days, so things like, coping in labour, induction of labour, and even virtual tours around the hospital, they have that in [my local Health Board] now, they have a link to a site for [a local] maternity unit, so using this kind of thing, like video clips. My clients are quite visual, so sometimes just directing them to a website that has a lot of written information, I don't really feel [its] appropriate for some clients. Sometimes, I'll do that and hope they read it, but I'll maybe do a bit of both, try and find more visual things that they can watch." [Family Nurse, W]*

A small proportion of family nurses also mentioned that their teams created Facebook, Twitter or Instagram accounts for the FNP service in their local Health Boards as a way of encouraging clients to connect with one another and reduce isolation.

*"[My team has] a FNP Facebook, Twitter and Instagram accounts, because again socially-wise, we want clients to connect with each other. We aren't totally at that stage yet, but at least we've now got the social media platforms as well. There's been a lot of good that actually has come out of last year, you know, with things that've been sitting in the pipeline for ages and it's actually moved on quickly because we're so desperate to have, a virtual system with these things. There has been some good that has come out of such a terrible year." [Family Nurse, W]*

**Programme activities, assessments, and observations.** Both survey and qualitative findings all showed a negative impact on the ability of family nurses to effectively conduct assessments and observations using telehealth. Inability to conduct a comprehensive assessment of the physical environment of clients and undertake vital measurements via remote consultation presented a barrier to most health professionals during COVID-19 [13]. Similarly, in our study, Fig 2 illustrates the survey findings and shows that the ability to conduct assessments of the home environment was most affected, with 98% of family nurses reporting this had been negatively impacted, overall. Observations of the child were perceived to be negatively impacted by 94% of family nurses and child health assessments were reported as being worse by 86% of family nurses. No family nurses reported improvement in any of the key assessments and observation domains.

The qualitative findings further explain the survey findings. Family nurses felt that remotely assessing home environments was extremely difficult, or 'impossible' in some cases. Key challenges related to being unable to identify who else might be present in a room with a client and being unable to detect the overall condition of a home as well as more subtle signals including family dynamics, body language, smells, and potential hazards. One of the key theories underpinning the FNP programme is human ecology theory, which recognises the importance role external or environmental factors play in parental and child development [18,19]. Therefore, inability for family nurses to undertake full assessment of the home environment is a challenge for delivering the FNP programme. For instance, some of the family nurses explained that whilst NHS Attend Anywhere and Near Me helped to some extent, it did not facilitate a complete assessment of the home environment.

*"… the fact that you can say that you actually saw the child has been incredibly helpful by Attend Anywhere. But in terms of a robust assessment of the home environment…and again all of that stuff about picking up cues if you're thinking about domestic violence and what's going on for a family, that has been very, very challenging, doing that by virtual means." [Family Nurse, W]*

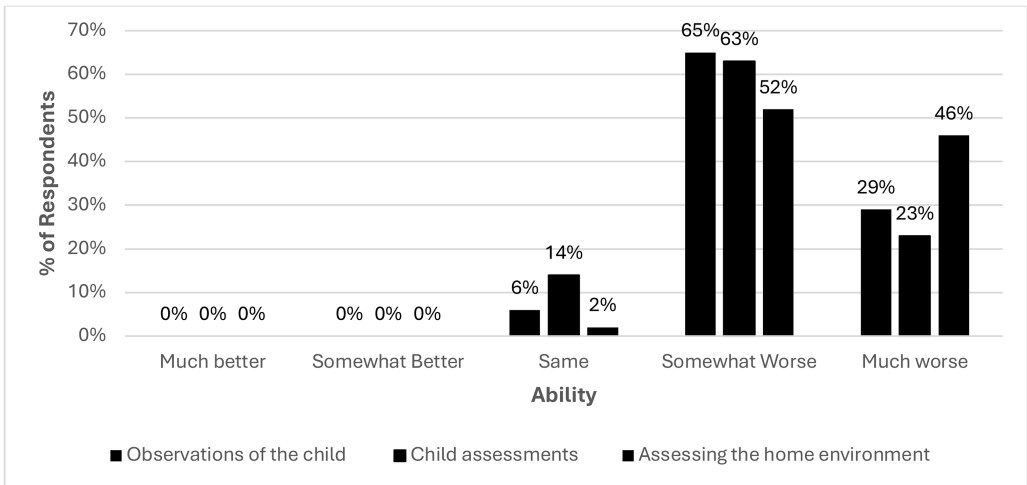

**Fig 2. Self-reported impact of telehealth on ability to carry out assessments and observations, including observations of the child and assessments of the home environment.**

**Challenges in the FNP delivery.** The study identified several challenges with the remote delivery of the FNP programme. Clients with vulnerabilities were deemed difficult to engage, therapeutic relationships were affected and issues with technology were recognised as some of the challenges.

**At-risk groups.** While many family nurses felt able to engage a proportion of their caseload in the programme remotely, particular challenges were reported for certain groups of clients such as those with complex vulnerabilities, migrants and clients who spoke English as a second language.

Clients living with mental health conditions such as ADHD, anxiety, and depression were perceived as having difficulties engaging with the programme remotely. Family nurses reported that clients with poor mental health were less comfortable to speak over the phone or by video call. They also noted an increase in poor mental health across their caseloads during the pandemic partially attributed to by the re-introduction of national lockdown restrictions. Some family nurses felt that clients struggling with their mental health were more likely to withdraw and hide their circumstances or avoid remote engagement attempts.

*"We can still deliver the programme and we can still check in with them, but they can shelter us a bit and hide from us if they don't want us to know, if they don't want to know that they're feeling particularly low or whatever, they just don't tell us. They're on the phone, and so it's easy for them to not have to admit if there's something going on, or something they're not coping with, or they're just feeling a bit rubbish that day, do you know." [Family Nurse, W]*

**Remote recruitment and therapeutic relationship.** During the COVID-19 outbreak, family nurses reported conducting recruitment predominantly by telephone. Despite recruitment rates remaining roughly the same, many family nurses reported that recruiting clients remotely using telehealth was considerably more challenging when compared with face-to-face recruitment.

The use of telehealth had negative impacts on therapeutic relationship building, communication and social interaction. Early relationship building was reported by family nurses to be the most significantly affected aspect, perceiving that it took longer and required more effort to establish a strong therapeutic relationship with clients remotely. The FNP programme is built on the expectation of establishing therapeutic a relationship between a family nurse and a client [18],

and ways to enhance this through remote delivery require attention. Family nurses felt eventually they were able to build therapeutic relationship with clients, but many reported that face-to-face contacts were key to establishing this more successfully.

*"So, I do feel like probably my relationship with the [clients] that I've recruited during lockdown is not quite as good as the ones I had recruited before. Because they've all been on videocall as opposed to face-to-face. So, I feel there is a bit of a barrier there." [Family Nurse, E]*

**Connectivity.** Connectivity issues were widely experienced by clients and staff using video call platforms such as Near Me. Audio and video issues were also common and many family nurses expressed that this made calls feel disjointed and frustrating. Other clients were reported to have more difficulties with technology and lower levels of literacy which was a challenging barrier for family nurses to overcome with clients. However, one family nurse reported that the provision of electronic devices as part of the adaptation to remote delivery provided opportunities to engage with clients.

*"I think that will be good for the [clients] because we can share that journey with them. 'Oh, brilliant, you've got a new iPad, let's talk through it together, let's build it up together and, oh, let's see if we can talk together on it.' I think that's a really positive thing and might help all the things that have been a bit of a struggle." [Family Nurse, W]*

**Access to resources, equipment and training.** Despite most family nurses reporting feeling well equipped to deliver the FNP remotely, a small proportion of family nurses felt that they had inadequate technology, devices or resources that impacted their abilities to deliver the programme.

Family nurses reported that their mobile phones and laptops were outdated and incompatible with certain software and some telehealth platforms.

*"Although we have [devices], it doesn't always work the way it should. And because there's been more of a reliance on a virtual way of working, that has been quite challenging. And I think, certainly in [my area], we've realised that a lot of our IT equipment is coming to the end of its life, but I think we're in a very long queue to have it replaced. So it's an ongoing issue and challenge". [Family Nurse, W]*

**Programme dosage and fidelity.** The FNP programme as a licensed programme has a clear structure and content, although it provides room for tailoring and adaptation to local context [18,20]. Adaptation and fidelity to remote delivery should therefore be balanced. Most clients did not experience any major differences in the number of contacts with family nurses during COVID-19 compared with prior programme delivery. Many clients reported keeping in contact with their family nurses via text messaging between scheduled programme contacts and most felt they were able to contact family nurses with queries and concerns during this time if necessary. The quote below from a client reflects this:

*"Every two weeks I think where we have a proper catchup, but I can text her or phone her any time if I've got any questions or anything. So, sometimes it's maybe like once a week, but it will just be over text type thing, but we have a proper catchup every two or three weeks." [Client (W), enrolled pre-COVID-19]*

Further analysis of the survey data conducted with family nurses showed that there were small differences in the over-all self-reported provision of visits to clients, before and during the COVID-19 pandemic. Fig 3 below shows that during COVID-19 there was a slight decrease in the number of contacts conducted. For example, family nurses reported a

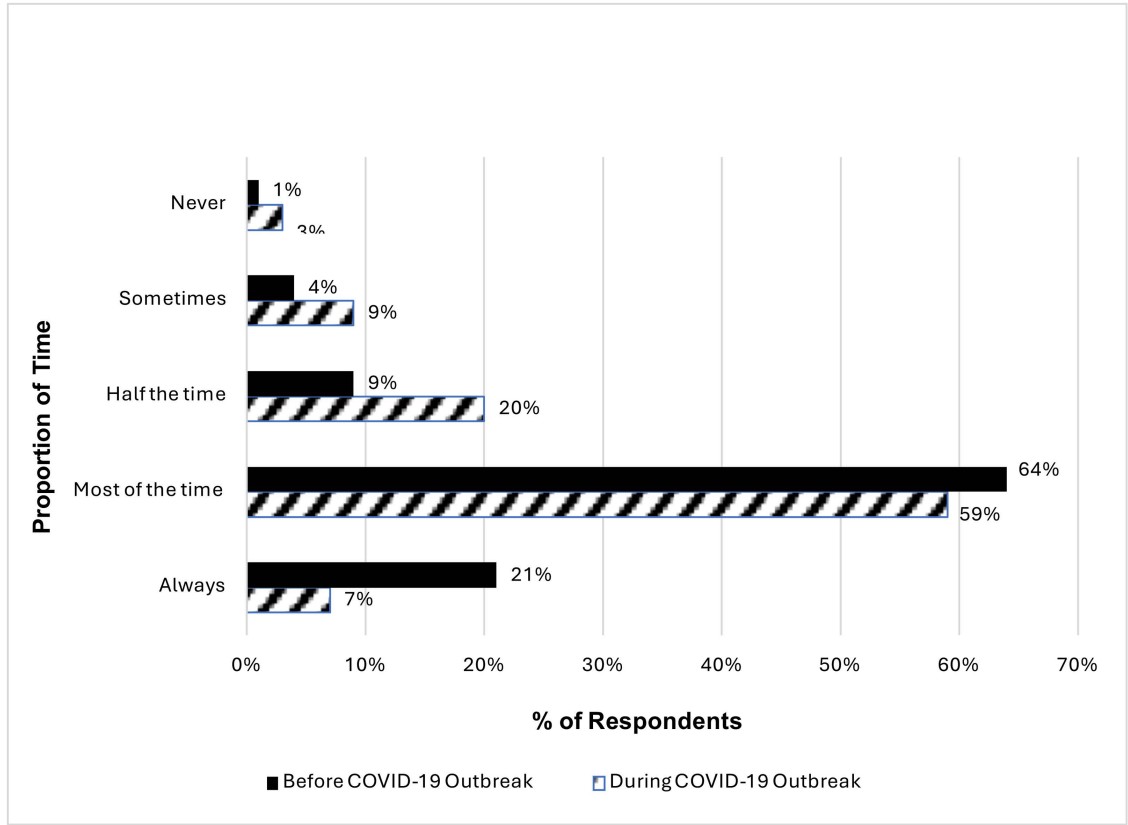

**Fig 3. Self-reported changes in the number of expected contacts, based on fidelity, provided to clients before and during the COVID-19 outbreak.**

decrease of 14% in always meeting programme fidelity following the outbreak, and a 5% decrease in meeting this most of the time.

## Training

Family nurses who deliver the FNP programme are highly experienced. However, the rapid adaptation of the programme for remote delivery, required training and many of the family nurses said they received training as illustrated by one of them below.

> "We had guidance for delivery of FNP on telehealth. We had a training session about the delivery of PIPE [Partners in Parenting education], activities via 'Near Me'." [Family nurse, N]

The survey data also indicated that a high number of nurses were trained to adapt the programme to remote delivery. Of surveyed family nurses, 90%, reported being provided with some form of guidance to support their work following COVID-19. In addition, 68% of family nurses reported being offered formal training opportunities to support remote working. Nevertheless, 23% reported being offered no training and 10% were unsure. Family nurses reported receiving or accessing training and support from a variety of sources. Local FNP teams were also cited as being

extremely supportive and involved in peer-learning activities, especially in the midst of frequent changes to COVID-19 guidelines.

## Future delivery of FNP

It was recognised that the adaptation of the FNP programme to remote delivery was an opportunity to explore the place of telehealth in the future delivery of the programme. Clients and family nurses were asked about their opinions on the future delivery of FNP based on their experiences of receiving or delivering the programme. Fifty-seven percent of surveyed family nurses agreed (46% somewhat agree and 11% strongly agree) that they would like to continue using telehealth to some extent to deliver FNP in the future; 29% disagreed (15% somewhat disagree and 14% strongly disagree) and 13% neither agreed nor disagreed.

When asked to rank most preferable delivery formats, 'in person visiting only', was most preferred by surveyed family nurses (55%). This was followed by 'mixed-mode (face-to-face and telehealth) delivery' (42%) with 'telehealth delivery only' being the least preferred option (77%) (see Fig 4 below).

When asked if FNP could be delivered effectively to clients using a multi-faceted (hybrid model) approach without compromising the essence or outcomes of the programme, 68% of family nurses agreed this was possible while 32% disagreed. Of the proportion of nurses who agreed with this statement, most commented that offering a hybrid model of delivery provided convenient options to maintain high levels of engagement for clients with commitments such as work or education, especially considering the target age group of the FNP clients are more prone to non-engagement [18]. However, family nurses emphasised that the provision of this should be dependent on clients' needs and their level of vulnerability.

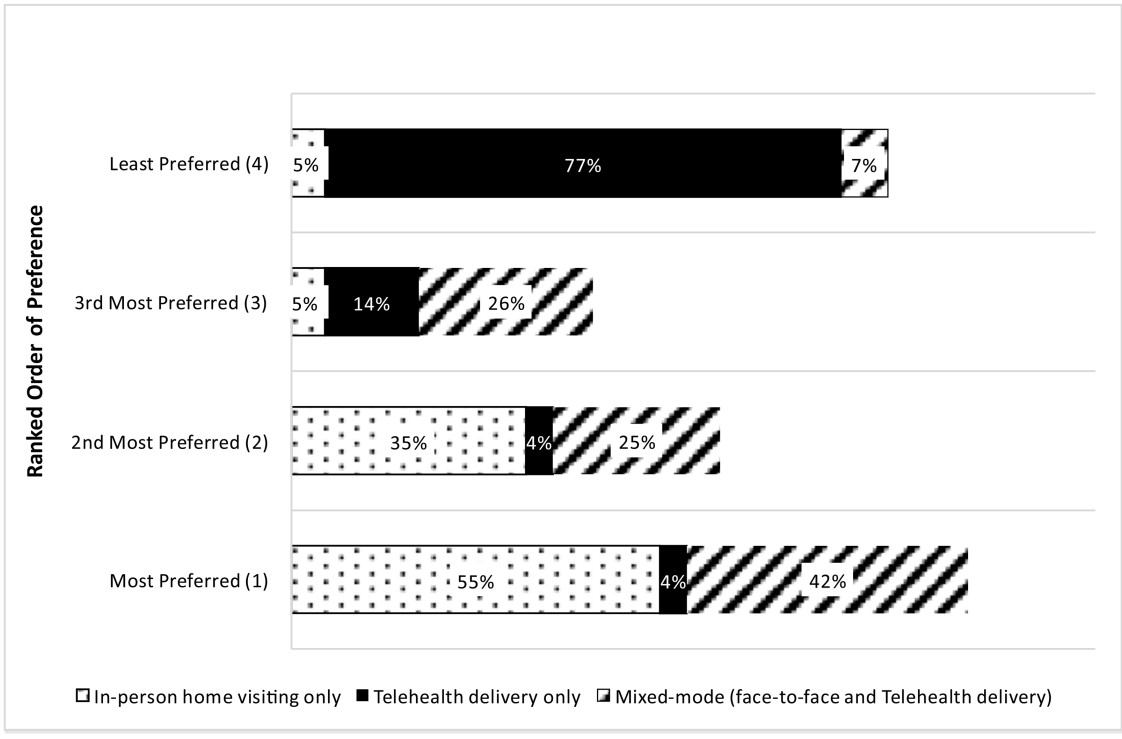

**Fig 4. Ranked order of preference for future delivery formats in FNP, reported by family nurses.**

The qualitative interview data further explained the findings of the survey data. It showed that a hybrid delivery model was beneficial for improved time management. Family nurses referred to benefits such as reduced work-related travel in between visits or meetings and limiting the time lost due to missed appointments. Improved access to clients living in more rural or remote areas was also perceived as a potential benefit of hybrid-delivery.

*"So, I'd normally do 50 to 100 miles a day and it was a lot of driving, so that would mean that when I got back from my last visit, maybe it would be nearly five o'clock, but then I'd still have to write up all the visits that I did that day and I didn't really have any time. So, I ended up doing at least an hour but probably more like two hours unpaid overtime a day, to be honest, because you just never had time to write everything up that you had to do."* [Family Nurse, E]

## Discussion

This study sought to understand the features of telehealth employed to deliver the FNP programme during COVID-19, how family nurses and clients responded to remote delivery of the programme and considered the wider implications of telehealth for the future delivery of FNP.

Prior to the outbreak of the COVID-19 in Scotland, the study found infrequent usage of telehealth in the FNP delivery, however as expected, this improved profoundly during the COVID-19 pandemic. The limited use of telehealth as part of FNP deliver prior to COVID-19 is similar to qualitative studies conducted in Australia that showed limited telehealth training, preparation and experiences of nurse-practitioners and primary health nurses prior to the COVID-19 pandemic [21,22]. However, the current study demonstrates that the addition of telehealth to nursing practice has the potential to enhance service delivery.

The findings indicated that some clients' preferred home visits, which the FNP programme was originally designed for. Home visiting fosters therapeutic relationship development between nurses and clients and ensures that the programme is able to capitalise on the delivery of the three key theories underpinning the FNP programme: social cognitive and self-efficacy, human ecology and attachment theories [18]. Home visiting enabled family nurses to perform and undertake effective physical assessment and examination of clients, a component that family nurses acknowledged was more challenging during remote delivery. This is reflected in the study of Halcomb et al. [23] in Australia where undertaking a physical examination and physical intervention presented a significant barrier to the remote delivery of services to clients. Nevertheless, there was no differences in the quality of mode of delivery as telehealth delivery was seen as mostly or somewhat the same as home visiting in terms of quality [23]. Our study was not designed to robustly assess the quality of telehealth delivery against home visits and hybrid delivery, although it was clear from our survey that 42% of family nurses preferred mixed-mode delivery of face-to-face and telehealth. Future studies could be designed to robustly examine the impact of the quality of mode of FNP delivery and how this influences outcomes across different client groups.

The FNP delivery during COVID-19 encountered several challenges including difficulty in recruiting clients remotely as compared with home visiting. This challenge was exacerbated with the recruitment of clients with complex vulnerabilities such as mental health conditions (including ADHD, anxiety, and depression), migrants and clients who spoke English as a second language. The findings show that in these instances clients faced difficulties when remotely engaging with programme. FNP clients could sometimes be more vulnerable [18] and it is important to assess which client groups would not be well suited to remote FNP delivery, for countries or organisation who may be considering a hybrid delivery of the FNP in the future.

The use of telehealth had negative effects on therapeutic relationship building between family nurses and clients. This was evident as clients were less able to disclose and communicate feelings about issues remotely over the phone or by videocall. This made it challenging for family nurses to pick up on subtle cues that would usually alert them to an issue during a face-to-face interaction. On the contrary, most participants or studies in the rapid review of Morrison et al. [5]

found telehealth had a positive impact on establishing therapeutic relationships between family nurses and clients. However, Hickey et al. [24] in their study to explore social care practitioners' experiences of delivering digital interventions to vulnerable children and families during the COVID-19 pandemic, found challenges in the therapeutic relationship between social care practitioners and clients in Ireland. The difference between Morrison el al. [5] and Hickey et al. [24] might well be the characteristics of the client groups involved, emphasising the importance of assessing the nature of client groups who are likely to respond positively to remote FNP delivery prior to the implementation of a hybrid FNP hybrid delivery programme. Also, it is understandable that it takes more time for the healthcare providers to establish relationship with the clients and may take a little longer for strong therapeutic relationships to develop remotely. Therefore implementation of any remote delivery component to an FNP programme would require careful monitoring to ensure there is no risk to the therapeutic relationship between family nurses and their clients.

Telehealth undoubtedly provided a practical and feasible alternative to home visiting during the pandemic. This was made possible via the use of technologies such as telephone calls, SMS (text) messaging, social media (WhatsApp, Facebook, YouTube, etc.), video calls via NHS Attend Anywhere and NHS Near Me to continually deliver services to clients. Telephone calls and SMS text messaging were the most widely used technologies in delivering the services remotely. This was corroborated in several studies during the COVID-19 pandemic [5,21,22]. Additionally, Halcomb et al. [23] found similar results where telephone was the mostly used method of delivering telehealth, and half (50%) of participants used a blend of telephone and video calls.

Also, connectivity issues such as lack of WiFi and internet networks and unavailability of technologies such as laptops coupled with low literacy levels of clients presented practical difficulties in remotely delivering the FNP programme. Although most family nurses were well equipped to deliver the FNP remotely, a small proportion felt that they had inadequate technology, devices or resources that impacted their abilities to deliver the programme. Family nurses from several Health Boards reported that their mobile phones and laptops were outdated and incompatible with certain software and telehealth platforms. Some devices did not have cameras, which made delivering video call contacts with clients and meetings challenging. These challenges with remote service delivery are reported in several other studies too [5,21,22,24].

In terms of programme dosage and fidelity, the study found that there was slight reduction in the number of contacts made with clients during the COVID-19 outbreak as compared to pre-COVID-19. However, telehealth offered several clients the opportunity to keep contact with their family nurses via text messaging between scheduled programme contacts and most felt they were able to contact family nurses with queries and concerns, if necessary. This is similar to the study by Mathew et al. [25] where there was a decline in the number of visits during the initial period of the COVID-19. The opportunity telehealth offers to clients to contact family nurses between scheduled programme contacts is however and interesting finding, which may require careful guidance in any remote delivery of an FNP programme in order to manage staff workload.

Considering views on the future delivery of the FNP programme, most clients preferred home visiting as compared to telehealth delivery. For family nurses, while some of them preferred some elements of telehealth for FNP delivery, it was recognised that the mode of delivery should largely depend on the client's needs and preferences. This is consistent with Halcomb et al.'s study [23], where primary healthcare professionals wanted the continued use of telehealth but with improvement and funding. Certainly, telehealth delivery requires investment in personnel and resources, but when fully embedded into a service, even in a hybrid model can offer opportunities to reach remote communities and hard-to-reach clients [5]. As home visiting is still the predominant delivery mode in the 11 mainland Health Boards in Scotland post-COVID-19, telehealth may present an opportunity to expand the FNP programme to Island Health Boards and more remote and rural areas in Scotland not yet delivering the programme and other similar areas globally. This is certainly one of the strengths of incorporating remote delivery to the FNP programme and decision makers may need to consider the value of investing in resources to promote remote delivery of aspects of the FNP programme, whilst monitoring the benefits clients derive from the programme.

## Limitations

Data collection for qualitative interviews, focus groups and surveys were undertaken remotely due to COVID-19. It is possible that this approach to data collection might have excluded the perspectives of the most vulnerable clients due to digital exclusion or other barriers preventing their participation. Recruitment of clients via their family nurse presents a problem of selection bias. There is a risk that the family nurses preselected clients who had a more favourable opinion and experience of the service. This could mean that clients with less favourable opinions or those from more vulnerable backgrounds could have been excluded from the sample. Demographic characteristics of the small number of participants who took part in the study was not presented to protect participants confidentiality. The survey was also purely exploratory, to support the qualitative findings and provide a broad snapshot to address the aims of the study, therefore no inferential statistics were undertaken.

Information governance requirements prevented demographics data from being captured, this information could have been used to further contextualise the findings.

## Conclusion

Despite the challenges of delivering the FNP remotely during COVID-19, both family nurses and clients expressed a positive view that telehealth could play some role in future delivery, for instance, to fill gaps in communication or follow-up on concerns. Telehealth was considered as a valuable addition to the delivery of FNP and investment in this could enhance care and support of children and families. Future research could be robustly designed to assess telehealth delivery against hybrid and home visits, and how the quality of different modes of delivery influence outcomes across different client groups. An economic evaluation of the value for money of different modes of delivery could also be insightful for decision makers.

## Acknowledgments

We would like to thank all participants who took part in the study. Also, we would like to thank Niamh Woodier, who contributed to the data collection and some aspects of the analysis.

## Author contributions

**Conceptualization:** Lawrence Doi.

**Data curation:** Lawrence Doi, Kathleen Morrison, Thomas Hughes.

**Formal analysis:** Lawrence Doi, Kathleen Morrison, Thomas Hughes.

**Funding acquisition:** Lawrence Doi.

**Investigation:** Lawrence Doi, Kathleen Morrison, Thomas Hughes.

**Methodology:** Lawrence Doi, Kathleen Morrison, Thomas Hughes.

**Project administration:** Kathleen Morrison.

**Supervision:** Lawrence Doi.

**Writing – original draft:** Lawrence Doi, Kathleen Morrison, Emmanuel Kwadwo Anago, Thomas Hughes.

**Writing – review & editing:** Lawrence Doi, Kathleen Morrison, Emmanuel Kwadwo Anago, Thomas Hughes.

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
