## [Decision Letter · Decision Letter 0]

9 Jul 2025

Dear Dr. Doi,

Thank you for submitting your manuscript to PLOS ONE. After careful consideration, we feel that it has merit but does not fully meet PLOS ONE’s publication criteria as it currently stands. Therefore, we invite you to submit a revised version of the manuscript that addresses the points raised during the review process.

We look forward to receiving your revised manuscript.

Kind regards,

Muhammad Abdul Rehman Rashid, PhD

Academic Editor

PLOS ONE

Journal Requirements:

“The evaluation was funded by a grant from the Scottish Government. The views expressed in this publication are those of the authors and not necessarily those of the funders. The funders had no role in study design, data collection and analysis, decision to publish, or preparation of the manuscript. We would like to thank Niamh Woodier, who contributed to the data collection and some aspects of the analysis in the original report to the funders.”

“The evaluation was funded by a grant (FNP 2020/21–Insights Research) from the Scottish Government. The views expressed in this publication are those of the authors and not necessarily those of the funders. The funders had no role in study design, data collection and analysis, decision to publish, or preparation of the manuscript.”

4. In the online submission form, you indicated that “However, data are available upon request for researchers who meet the criteria for access to confidential data from the first author or through the Ethics Committees involved.”

Reviewers' comments:

Reviewer's Responses to Questions

**Comments to the Author**

1. Is the manuscript technically sound, and do the data support the conclusions?

Reviewer #1: Yes

Reviewer #2: Yes

Reviewer #3: Partly

Reviewer #4: Yes

2. Has the statistical analysis been performed appropriately and rigorously?

Reviewer #1: Yes

Reviewer #2: Yes

Reviewer #3: N/A

Reviewer #4: I Don't Know

3. Have the authors made all data underlying the findings in their manuscript fully available?

Reviewer #1: No

Reviewer #2: Yes

Reviewer #3: No

Reviewer #4: No

4. Is the manuscript presented in an intelligible fashion and written in standard English?

Reviewer #1: Yes

Reviewer #2: Yes

Reviewer #3: Yes

Reviewer #4: Yes

Reviewer #1: The study "Evaluation of the Delivery of the Family Nurse Partnership Program in Scotland

during the COVID-19 Pandemic" provides valuable insights for the role of telehealth in nursing. Study is well structured and using quantitative and qualitative data make it more fruitful for community. Practical future recommendations are also worthy. Proof reading is required for grammatical errors and data should be available for more clarity.

Reviewer #2: All things considered, this paper significantly advances the assessment of the Family Nurse Partnership (FNP)'s implementation in Scotland during the COVID-19 pandemic, especially with regard to the telehealth service adaptation. To raise the manuscript's level of scientific and methodological quality, a number of important elements still need to be reinforced. The authors must include a thorough description of the study methodologies, including the approach used, the data collection strategies employed, and the sort of analysis done, in the abstract section. Furthermore, the study's results and first conclusions have not been made clear. It is important to highlight key findings so that readers are aware of the study's primary accomplishments right once. The author has given a thorough historical and geographical overview of the FNP implementation in the introduction. The research gap that underlies the necessity of this study, as well as the novelty or innovation this study offers in comparison to earlier studies conducted in other nations, such the UK, Canada, and the Netherlands, require a more thorough explanation. To make the research objectives more targeted and focused on the research problems that need to be addressed, they must also be defined, both generally and operationally. Although the author refers to a mixed-techniques approach in the methods section, no specific explanation of the design—such as convergent, explanatory sequential, or embedded design—has been provided. The reasoning for this method's application should be described, as well as the steps taken to integrate qualitative and quantitative data. Furthermore, a more thorough description of the data analysis methods should be included for both qualitative and quantitative data. This includes the software utilized, any validity or reliability standards, and the interpretation of the results. To help the reader comprehend the findings, the author has categorized them into major themes in the results and discussion part. However, the explanation of the results is typically descriptive and lacks a thorough analysis. The significance of the discovered data must be explained in greater depth, along with how it relates to the pertinent theory and context. In order for readers to fully understand the dynamics of the findings, the percentage of quantitative data that is supplied through surveys still uses broad language; instead, it needs to be extended in more narrative and specific terms. In order to enhance the argument and create a connection between the facts and the analytical narrative, data visualization tools like tables and graphs should be used in conjunction with more in-depth interpretations rather than just as visual aids. The author should also explain how these findings affect the development or reinforcement of the theory that underpins the FNP program, especially the theory of therapeutic partnerships in community health care. A few more phrases are required to clearly clarify the primary findings of this study, even though the author has indicated in the conclusion that telehealth has the potential to be employed in a hybrid fashion in the future when FNP is implemented. In order to provide more substantial value to the creation of community-based nursing care policies, the conclusion should include cover scientific contributions, real-world applications, and future research prospects. To be published in a respectable journal, this manuscript still needs to be improved in terms of goal formulation, data analysis depth, and methodological clarity, but overall it has the potential to be a significant contribution to the conversation on modifying maternal and child health services during the pandemic.

Reviewer #3: Overall assessment

This study addresses an important question: how the Family Nurse Partnership (FNP) in Scotland adjusted to COVID‑19 restrictions and what that meant for service delivery. Although the topic is unquestionably relevant to public‑health practice, the manuscript reads more like an internal service report than a research article that meets PLOS ONE standards. The results are mostly descriptive, several methodological details are missing, and the paper does not yet comply with the journal’s data‑sharing policy. Significant revision is required before the work can be considered for publication.

Major comments

Sample size and representativeness. Only fifteen client or family members were interviewed against thirty‑one nurses, and the manuscript gives no information on the number of eligible families, the proportion approached, or the reasons for non‑participation. In addition, the study covers eleven Scottish Health Boards, but the authors do not state how many Boards exist in total. Without these denominators it is impossible to judge representativeness or assess selection bias.

Research questions. The introduction reviews international literature but concludes with vague aims such as “evaluate” and “understand challenges”. Explicit research questions would give the study sharper focus.

Quantitative component. The survey results are presented descriptively, without confidence intervals, hypothesis tests or any justification for the sample size. If inferential statistics are inappropriate, the authors need to explain why the survey was undertaken and clarify that its role is purely exploratory.

Integration of methods. Qualitative and quantitative findings appear in parallel blocks and seldom interact. The paper would benefit from explicitly linking survey patterns.

Rigour of the thematic analysis. There is no mention of an external audit of the coding framework, no assessment of inter‑coder reliability and no discussion of how emergent codes were incorporated. Greater transparency is needed to demonstrate analytical rigour.

Numerical detail. Key programme metrics, such as the reduction in home visits or the proportion of contacts delivered remotely, are discussed qualitatively but never quantified. The abstract in particular contains no numbers, making it difficult for readers to gauge the magnitude of change.

Statements on telehealth. The conclusion that telehealth “could play some role” is vague. The authors should offer concrete, operational recommendations—for example, criteria for hybrid scheduling or digital‑access assessments.

Data availability. The data are available only on request because of potential identifiability. This arrangement conflicts with PLOS ONE’s open‑data policy. At minimum, anonymised survey aggregates and redacted transcripts should be deposited in a public repository, or the authors must seek an exemption from the editors.

Timeliness. Data collection finished in 2021, yet the manuscript is being submitted four years later. The authors should explain why the findings remain current and describe any subsequent changes in FNP practice that might influence interpretation.

Minor comments

Demographic description. Governance rules may forbid disclosing individual‑level data, but an aggregated table of age bands, parity and deprivation quintiles is essential for readers to assess transferability.

Selection bias. Clients were recruited via their own nurse, a process likely to introduce desirability bias. The limitation deserves a fuller discussion.

Typographical error. At line 63 the phrase reads “programme’s primary objectives of the programme”. The duplication should be removed.

Reviewer #4: 1. The results are detailed and robust; however, including a brief summary of the main themes under each sub-section would enhance clarity. This would help readers follow the flow of the discussion more easily and better understand the similarities and differences between the findings from the qualitative interviews and the survey.

2. Under the methods section, ensure that the missing components of COREQ (Consolidated Criteria for Reporting Qualitative Research) are provided. E.g. positionality.

3. Some grammatical errors are noticeable in the text. Kindly edit as appropriate. Few examples are on lines 28, 80, 101, 112, 114, 166.

**Do you want your identity to be public for this peer review?** For information about this choice, including consent withdrawal, please see our Privacy Policy

Reviewer #1: **Yes: ** Hafiz Muhammad Zakria

Reviewer #2: **Yes: ** Alimin Alwi

Reviewer #3: No

Reviewer #4: No

---

## [Author Response · Author response to Decision Letter 1]

29 Aug 2025

Comments by reviewers and our responses:

Reviewer #1

1. The study "Evaluation of the Delivery of the Family Nurse Partnership Program in Scotland during the COVID-19 Pandemic" provides valuable insights for the role of telehealth in nursing. Study is well structured and using quantitative and qualitative data make it more fruitful for community. Practical future recommendations are also worthy. Thank you for this comment.

2. Proof reading is required for grammatical errors and data should be available for more clarity. We have now done a thorough proofreading of the manuscript. Unfortunately, NHS ethics requirement does not permit the sharing of our research data publicly, but request could be made through the ethics committee (phs.PBPP@phs.scot). We have sought exemption from the Editor.

Reviewer #2:

1. All things considered, this paper significantly advances the assessment of the Family Nurse Partnership (FNP)'s implementation in Scotland during the COVID-19 pandemic, especially with regard to the telehealth service adaptation. To raise the manuscript's level of scientific and methodological quality, a number of important elements still need to be reinforced. Thank you for this comment.

2. The authors must include a thorough description of the study methodologies, including the approach used, the data collection strategies employed, and the sort of analysis done, in the abstract section. Furthermore, the study's results and first conclusions have not been made clear. It is important to highlight key findings so that readers are aware of the study's primary accomplishments right once. We enhanced the description of the study methods and made the findings and conclusions clearer in the abstract.

3. The author has given a thorough historical and geographical overview of the FNP implementation in the introduction. The research gap that underlies the necessity of this study, as well as the novelty or innovation this study offers in comparison to earlier studies conducted in other nations, such the UK, Canada, and the Netherlands, require a more thorough explanation. To make the research objectives more targeted and focused on the research problems that need to be addressed, they must also be defined, both generally and operationally. We have provided further explanation of the research gap that our study addressed in relation to previous studies. We have also clearly articulated the aims of the study in both the abstract and at the end of the introduction section.

4. Although the author refers to a mixed-techniques approach in the methods section, no specific explanation of the design—such as convergent, explanatory sequential, or embedded design—has been provided. The reasoning for this method's application should be described, as well as the steps taken to integrate qualitative and quantitative data. We have clarified in the methods (and also in the abstract) that we used mixed-methods parallel design and explained the rationale for using this design. We have also explained the approach we used to integrate qualitative and quantitative data.

5. Furthermore, a more thorough description of the data analysis methods should be included for both qualitative and quantitative data. This includes the software utilized, any validity or reliability standards, and the interpretation of the results. To help the reader comprehend the findings, the author has categorized them into major themes in the results and discussion part. However, the explanation of the results is typically descriptive and lacks a thorough analysis. The significance of the discovered data must be explained in greater depth, along with how it relates to the pertinent theory and context. We have provided further description of the data analysis methods. We have also been explicit about the NVivo software and MS excel package we used to facilitate the data analysis. We have clarified how the survey questionnaire was developed and been explicit that no validated instrument was used.

We have now revised the findings and demonstrated how our findings link to pertinent theories and context where possible.

6. In order for readers to fully understand the dynamics of the findings, the percentage of quantitative data that is supplied through surveys still uses broad language; instead, it needs to be extended in more narrative and specific terms. In order to enhance the argument and create a connection between the facts and the analytical narrative, data visualization tools like tables and graphs should be used in conjunction with more in-depth interpretations rather than just as visual aids. We have now revised the findings and made the link between the interview and survey data more findings more explicit. We have included two tables and four figures to aid data visualisation, and in-depth interpretations added where possible.

7. The author should also explain how these findings affect the development or reinforcement of the theory that underpins the FNP program, especially the theory of therapeutic partnerships in community health care. We have discussed both in the findings and discussion sections how our findings relate to therapeutic relationship and other theories pertinent to the FNP programme.

8. A few more phrases are required to clearly clarify the primary findings of this study, even though the author has indicated in the conclusion that telehealth has the potential to be employed in a hybrid fashion in the future when FNP is implemented. In order to provide more substantial value to the creation of community-based nursing care policies, the conclusion should include cover scientific contributions, real-world applications, and future research prospects. The abstract and conclusion have been revised to clearly outline the scientific contributions, real-world applications and future research prospect.

9. To be published in a respectable journal, this manuscript still needs to be improved in terms of goal formulation, data analysis depth, and methodological clarity, but overall it has the potential to be a significant contribution to the conversation on modifying maternal and child health services during the pandemic. We hope the amendments we have made to the various sections address this point.

Reviewer #3:

Overall assessment

1. This study addresses an important question: how the Family Nurse Partnership (FNP) in Scotland adjusted to COVID 19 restrictions and what that meant for service delivery. Although the topic is unquestionably relevant to public health practice, the manuscript reads more like an internal service report than a research article that meets PLOS ONE standards. The results are mostly descriptive, several methodological details are missing, and the paper does not yet comply with the journal’s data sharing policy. Significant revision is required before the work can be considered for publication.

Thank you for your overall assessment of our paper. We hope the amendments we have made to the paper based on your specific comments below and the other comments from the other reviews have addressed this point.

Major comments

2. Sample size and representativeness. Only fifteen client or family members were interviewed against thirty one nurses, and the manuscript gives no information on the number of eligible families, the proportion approached, or the reasons for non participation. In addition, the study covers eleven Scottish Health Boards, but the authors do not state how many Boards exist in total. Without these denominators it is impossible to judge representativeness or assess selection bias.

We have now included the number of eligible clients at the time of data collection under ‘sample’ in the methods. As we have indicated in the limitations, clients were recruited through their family nurses and ethics requirements prevented the research team from having direct contacts with clients, therefore it is difficult to say the reasons for non-participation. We have also clarified under ‘design, setting and context’ that we recruited from all the 11 health boards (out of a total of 14) currently running the FNP programme in Scotland.

3. Research questions. The introduction reviews international literature but concludes with vague aims such as “evaluate” and “understand challenges”. Explicit research questions would give the study sharper focus.

Thank you. We have now clearly articulated the three main aims of the study.

4. Quantitative component. The survey results are presented descriptively, without confidence intervals, hypothesis tests or any justification for the sample size. If inferential statistics are inappropriate, the authors need to explain why the survey was undertaken and clarify that its role is purely exploratory. The survey was purely exploratory to support the qualitative results, and no inferential statistics were undertaken.

5. Integration of methods. Qualitative and quantitative findings appear in parallel blocks and seldom interact. The paper would benefit from explicitly linking survey patterns. We have revised the findings considerably and have explicitly linked the qualitative and survey findings.

6. Rigour of the thematic analysis. There is no mention of an external audit of the coding framework, no assessment of inter coder reliability and no discussion of how emergent codes were incorporated. Greater transparency is needed to demonstrate analytical rigour. We have provided further clarity of the analytical process and also added a paragraph on positionality as requested by reviewer number 4

7. Numerical detail. Key programme metrics, such as the reduction in home visits or the proportion of contacts delivered remotely, are discussed qualitatively but never quantified. The abstract in particular contains no numbers, making it difficult for readers to gauge the magnitude of change. We have revised the findings considerably and where relevant we have provided figures to support the qualitative findings. We have also included a key survey result in the abstract.

8. Statements on telehealth. The conclusion that telehealth “could play some role” is vague. The authors should offer concrete, operational recommendations—for example, criteria for hybrid scheduling or digital access assessments. Thank you for this helpful comments. We have now revised and strengthened the conclusion.

9. Data availability. The data are available only on request because of potential identifiability. This arrangement conflicts with PLOS ONE’s open data policy. At minimum, anonymised survey aggregates and redacted transcripts should be deposited in a public repository, or the authors must seek an exemption from the editors. Unfortunately, NHS ethics requirement does not permit the sharing of our research data publicly, but request could be made through the ethics committee (phs.PBPP@phs.scot). We have sought exemption from the Editor.

10. Timeliness. Data collection finished in 2021, yet the manuscript is being submitted four years later. The authors should explain why the findings remain current and describe any subsequent changes in FNP practice that might influence interpretation. Unfortunately due to staff leaving posts the paper could not submitted earlier than we had hoped. However, we believe the findings are relevant to the delivery of the FNP programme post COVID-19. We have also included the following statement in the discussion section to describe current FNP practice in Scotland.

“As home visiting is still the predominant delivery mode in the 11 mainland Health Boards in Scotland post-COVID-19, telehealth may present an opportunity to expand the FNP programme to Island Health Boards and more remote and rural areas in Scotland not yet delivering the programme and other similar areas globally.

Minor comments

11. Demographic description. Governance rules may forbid disclosing individual level data, but an aggregated table of age bands, parity and deprivation quintiles is essential for readers to assess transferability. We appreciate that demographic information would have enhanced the manuscript further. However, as we have noted in the limitations, information governance requirements prevented demographics data and deprivation data from being captured from the 15 clients that took part in the study.

12. Selection bias. Clients were recruited via their own nurse, a process likely to introduce desirability bias. The limitation deserves a fuller discussion. Addressed. We have now discussed desirability bias and recruitment in under the limitations.

13. Typographical error. At line 63 the phrase reads “programme’s primary objectives of the programme”. The duplication should be removed. Apologies for the oversight. Change now made.

Reviewer #4:

1. The results are detailed and robust; however, including a brief summary of the main themes under each sub-section would enhance clarity. This would help readers follow the flow of the discussion more easily and better understand the similarities and differences between the findings from the qualitative interviews and the survey. Thank you. We have revised the findings considerably and we believe this comment has now been addressed.

2. Under the methods section, ensure that the missing components of COREQ (Consolidated Criteria for Reporting Qualitative Research) are provided. E.g. positionality. Addressed. We’ve added an additional paragraph about positionality in the methods.

3. Some grammatical errors are noticeable in the text. Kindly edit as appropriate. Few examples are on lines 28, 80, 101, 112, 114, 166. Thank you. These have now been addressed.

Additional reviewers’ comments from attachment

Abstract

- An explanation of the methods used, results, and conclusions needs to be added

- Clarify the methods used in this study

- The research findings need to be emphasized

Thank you. We have revised the abstract considerably and we believe this comment has now been addressed.

Introduction

- A detailed explanation of the research gap and the novelty of the research

- Clarify the objectives of the research

We have provided explanation of the research gap and clearly articulated the research aims under the introduction.

Methods

- Clarify the design of the methods

- Provide a detailed explanation of the data analysis used in this study

Thank you. We have revised the methods considerably and we believe this comment has now been addressed.

Research Findings and Discussion

- A more in-depth analysis of the data findings

- Present specific percentages and also present them in narrative form

- It is permissible to include more in-depth visual interpretations

- Explain the implications of the theory

Thank you. We have revised the findings and discussion sections considerably and we believe these points have now been addressed.

Conclusion

- Add a few sentences explaining the research findings We have revised the conclusions and we believe this comment has now been addressed.

---

## [Decision Letter · Decision Letter 1]

18 Sep 2025

Dear Dr. Doi,

We look forward to receiving your revised manuscript.

Kind regards,

Muhammad Abdul Rehman Rashid, PhD

Academic Editor

PLOS ONE

Journal Requirements:

Reviewers' comments:

Reviewer's Responses to Questions

**Comments to the Author**

Reviewer #1: (No Response)

Reviewer #2: (No Response)

Reviewer #3: (No Response)

Reviewer #4: (No Response)

2. Is the manuscript technically sound, and do the data support the conclusions?

Reviewer #1: Yes

Reviewer #2: Yes

Reviewer #3: Yes

Reviewer #4: Yes

3. Has the statistical analysis been performed appropriately and rigorously?

Reviewer #1: Yes

Reviewer #2: Yes

Reviewer #3: N/A

Reviewer #4: Yes

4. Have the authors made all data underlying the findings in their manuscript fully available?

Reviewer #1: Yes

Reviewer #2: Yes

Reviewer #3: No

Reviewer #4: Yes

5. Is the manuscript presented in an intelligible fashion and written in standard English?

Reviewer #1: Yes

Reviewer #2: Yes

Reviewer #3: Yes

Reviewer #4: Yes

Reviewer #1: Overall Assessment

This is a relevant and timely study on the delivery of FNP during COVID-19. The design of mixed methods adds strength, but the document still needs improvements in clarity, methodological transparency and analytical depth to comply with PLOS One standards.

Major Comments

Methods: Provide more details about client recruitment, representativeness and how qualitative/quantitative data were integrated.

Analysis: Add clarity on thematic coding and justify why only descriptive survey data is used. Include confidence intervals if possible.

Discussion: Go beyond the description: link findings to the theory, highlight the novelty compared to other countries and offer concrete recommendations for hybrid delivery.

Data Availability: The current statement conflicts with the open data policy of PLOS One; Consider depositing anonymized aggregates or drafted transcripts.

Minor Comments

Summary: Add sample sizes and a key numerical result; sharpen the conclusion with contribution/relevance of the policy.

Introduction: Shorten background; emphasize the research gap.

Language: Necessary the final review; Avoid vague terms such as "many" or "several."

Conclusion: Provide clearer political implications and describe scientific contributions

Recommendation

Strong potential, but needs clearer methods, deeper analysis and better alignment with the PLOS One requirements.

Reviewer #2: The revised manuscript has significantly improved and now meets publication requirements. By clarifying the study design, strengthening methodological details, and increasing the depth of data analysis, the authors have addressed the concerns of previous reviewers. The inclusion of tables and figures, as well as a clearer connection between qualitative and quantitative results, has improved readability and interpretation. The study results are linked to relevant theory and practical implications for future family care provision. Although data are still limited due to ethical requirements, the explanations provided are plausible. Overall, the paper is well-written, technically sound, and provides important information on telehealth adaptations.

Reviewer #3: I appreciate the extensive revisions made to the manuscript. The authors have clearly addressed most of the concerns raised in the first review, and the paper is now much stronger in terms of methodological transparency, integration of findings, and clarity of conclusions. The inclusion of explicit research questions, numerical detail, and discussion of limitations all represent significant improvements.

While I understand the governance restrictions, it would still be valuable for readers to see at least an aggregated table of client characteristics (e.g., age bands, parity, deprivation quintiles). Such information is essential for assessing the transferability of findings. If this is not possible, the limitation should be underscored more strongly in the manuscript.

Reviewer #4: Most of the comments have been addressed, though some are still not met. Such as:

4. Justify replacing missing Gini data with the country average. Is this the standard practice or what is the rationale?

DISCUSSION

5. In some paragraphs, authors need to contextualize the findings and draw adequate implications of the results. Also ensure to provide new interpretation of the study findings and not a repetition of the literature review in the previous sections of the manuscript.

**Do you want your identity to be public for this peer review?** For information about this choice, including consent withdrawal, please see our Privacy Policy

Reviewer #1: No

Reviewer #2: No

Reviewer #3: No

Reviewer #4: No

---

## [Author Response · Author response to Decision Letter 2]

30 Oct 2025

Reviewer #1 Response

1. Overall Assessment

This is a relevant and timely study on the delivery of FNP during COVID-19. The design of mixed methods adds strength, but the document still needs improvements in clarity, methodological transparency and analytical depth to comply with PLOS One standards.

Thank you.

2. Major Comments

Methods: Provide more details about client recruitment, representativeness and how qualitative/quantitative data were integrated.

The recruitment and data collection under the methods, provide adequate details of how family nurses and clients were recruited. We have also added that the 90 nurses who took part in the survey represent a response rate of 41% across Scotland. We have also further explained the mixed-methods parallel (convergent) designed we employed, including how data were integrated.

3. Analysis: Add clarity on thematic coding and justify why only descriptive survey data is used. Include confidence intervals if possible.

We have clarified the thematic analysis process we used.

The survey was purely exploratory, to support the qualitative findings and provide a broad snapshot to address the aims of the study, which were 1) to understand the features of telehealth employed to deliver the FNP programme during COVID-19 in Scotland; 2) to examine how FNP nurses and clients responded to the delivery of FNP through telehealth; 3) to evaluate the challenges of delivering the FNP through telehealth during COVID-19 and its implications for future delivery of the programme.

We have added the exploratory nature of the quantitative component as a limitation within the manuscript.

4. Discussion: Go beyond the description: link findings to the theory, highlight the novelty compared to other countries and offer concrete recommendations for hybrid delivery.

In the previous round of revisions, we significantly revamped the discussion, linking it to theories underpinning the FNP. Point number 3 from reviewer #2 and point number 1 from reviewer #3 duly acknowledged this. However, we have further strengthened the discussion section to address the reviewer’s point.

5. Data Availability: The current statement conflicts with the open data policy of PLOS One; Consider depositing anonymized aggregates or drafted transcripts.

Unfortunately, NHS ethics requirement does not permit the sharing of our research data publicly. Reviewer #2 also highlights that this explanation is plausible. We have also sought exemption from the Editor.

6. Minor Comments

Summary: Add sample sizes and a key numerical result; sharpen the conclusion with contribution/relevance of the policy.

The abstract clearly has sample sizes, and we have refined the abstract based on the previous reviewers’ comments. We have added a recommendation for policy.

7. Introduction: Shorten background; emphasize the research gap.

Thanks for your comments. The introduction was revised based on comments from other reviewers and we strongly believe that shortening this at this stage will not fully address comments raised by other reviewers. The introduction clearly emphasizes the research gap that this study set out to address.

8. Language: Necessary the final review; Avoid vague terms such as "many" or "several."

The use of terms such as “many” and “several” is commonly used to present qualitative findings as the use of numbers is generally not considered a good practice in qualitative e research because it misrepresents the study's goal of in-depth understanding, implies misleading generalizability from small samples, strips context, and creates a false impression of precision for subjective experiences.

9. Conclusion: Provide clearer political implications and describe scientific contributions

Throughout the discussion and conclusion sections, we have highlighted the scientific contributions of the study as well as policy implications.

10. Recommendation

Strong potential, but needs clearer methods, deeper analysis and better alignment with the PLOS One requirements.

Thank you. We believe the revisions we have made address the second part of this point.

Reviewer #2

1. The revised manuscript has significantly improved and now meets publication requirements. By clarifying the study design, strengthening methodological details, and increasing the depth of data analysis, the authors have addressed the concerns of previous reviewers.

Thank you.

2. The inclusion of tables and figures, as well as a clearer connection between qualitative and quantitative results, has improved readability and interpretation.

Thank you.

3. The study results are linked to relevant theory and practical implications for future family care provision.

Thank you.

4. Although data are still limited due to ethical requirements, the explanations provided are plausible.

Thank you.

5. Overall, the paper is well-written, technically sound, and provides important information on telehealth adaptations.

Thank you.

Reviewer #3:

1. I appreciate the extensive revisions made to the manuscript. The authors have clearly addressed most of the concerns raised in the first review, and the paper is now much stronger in terms of methodological transparency, integration of findings, and clarity of conclusions. The inclusion of explicit research questions, numerical detail, and discussion of limitations all represent significant improvements.

Thank you.

2. While I understand the governance restrictions, it would still be valuable for readers to see at least an aggregated table of client characteristics (e.g., age bands, parity, deprivation quintiles). Such information is essential for assessing the transferability of findings. If this is not possible, the limitation should be underscored more strongly in the manuscript.

Thanks for your comment. Unfortunately, we didn’t collect demographic data of the 15 clients who took part in interviews to protect participants confidentiality because of the small sample involved. Also, the goal was to focus on the experiences of clients receiving the FNP programme. We have acknowledged this as a limitation of the study in the manuscript.

Reviewer #4:

1. Most of the comments have been addressed, though some are still not met. Such as:

Justify replacing missing Gini data with the country average. Is this the standard practice or what is the rationale?

Thank you. We are pleased we addressed most of your comments.

We didn’t present any Gini data so we are unsure what this comment is referring to.

2. DISCUSSION

In some paragraphs, authors need to contextualize the findings and draw adequate implications of the results. Also ensure to provide new interpretation of the study findings and not a repetition of the literature review in the previous sections of the manuscript.

In the previous round of revisions, we significantly revamped the discussion, linking it to theories underpinning the FNP. Point number 3 from reviewer #2 and point number 1 from reviewer #3 duly acknowledged this. However, we have further strengthened the discussion section to address the reviewer’s point.

Additional reviewer’s comment from attachment

Abstract

The abstract already explains the background, methods, results, and conclusions, but it would be stronger if it added key figures briefly to emphasize the magnitude of the findings.

Thanks for your comment. This comment was also raised in the previous round of revisions, and we revamped the abstract to include key figure from our findings. As this is a mixed methods study, we believe the abstract reflects our key findings.

Introduction

The introduction has outlined the context of the FNP and reviewed previous research, but it would have been better if the emphasis on the research gap had been made more acute.

Thanks for your comments. The introduction was revised based on comments from other reviewers in the previous round of revisions, providing a clear research gap that our study was designed to address. Please see rationale and aims of the study under the introduction.

Methods

The methods section already explains the design, data collection techniques, and analysis, but it would be stronger if the explanation of the justification for the choice of method was expanded.

We have provided rationale for the choice of the methods we used where relevant.

Research Findings and Discussion

The results section already presents qualitative and quantitative data with tables and visualizations, but it would be stronger if the interpretive analysis were expanded.

Thank you for your comment. In the previous round of revisions, we significantly revamped the findings and discussion. Point number 3 from reviewer #2 and point number 1 from reviewer #3 duly acknowledged this. However, we have further strengthened the discussion section to address the reviewer’s point.

Conclusion

The conclusion already confirms the potential of telehealth in the FNP hybrid model, but it would be stronger if it added explicit statements regarding scientific contributions and practical implications.

Throughout the discussion and conclusion sections, we have highlighted the scientific contributions of the study as well as policy implications.

---

## [Editor Report · Decision Letter 2]

5 Nov 2025

Evaluation of the delivery of the Family Nurse Partnership programme in Scotland during the COVID-19 pandemic

PONE-D-25-02060R2

Dear Dr. Doi,

We’re pleased to inform you that your manuscript has been judged scientifically suitable for publication and will be formally accepted for publication once it meets all outstanding technical requirements.

Kind regards,

Muhammad Abdul Rehman Rashid, PhD

Academic Editor

PLOS ONE
---

## [Editor Report · Acceptance letter]

PONE-D-25-02060R2

PLOS ONE

Dear Dr. Doi,

I'm pleased to inform you that your manuscript has been deemed suitable for publication in PLOS ONE. Congratulations! Your manuscript is now being handed over to our production team.

Kind regards,

on behalf of

Dr. Muhammad Abdul Rehman Rashid

Academic Editor

PLOS ONE